# An Outbreak of Subclinical Mastitis in a Dairy Herd Caused by a Novel *Streptococcus canis* Sequence Type (ST55)

**DOI:** 10.3390/ani11020550

**Published:** 2021-02-20

**Authors:** Cassandra Eibl, Martina Baumgartner, Verena Urbantke, Michael Sigmund, Katharina Lichtmannsperger, Thomas Wittek, Joachim Spergser

**Affiliations:** 1University Clinic for Ruminants, Department for Farm Animals and Veterinary Public Health, University of Veterinary Medicine Vienna, Veterinärplatz 1, 1210 Vienna, Austria; martina.baumgartner@vetmeduni.ac.at (M.B.); verena.urbantke@vetmeduni.ac.at (V.U.); michael.sigmund@vetmeduni.ac.at (M.S.); katharina.lichtmannsperger@vetmeduni.ac.at (K.L.); thomas.wittek@vetmeduni.ac.at (T.W.); 2Institute of Microbiology, University of Veterinary Medicine Vienna, Veterinärplatz 1, 1210 Vienna, Austria; joachim.spergser@vetmeduni.ac.at

**Keywords:** β-hemolytic streptococci, mastitis, multilocus sequence typing, *Streptococcus canis* ST55

## Abstract

**Simple Summary:**

Although *Streptococcus (S). canis* is mainly isolated from carnivores, intramammary infection in dairy cows caused by this bacterium have been reported. Cats and dogs with access to the barn have been considered as the main source of these infections. Here, we report subclinical mastitis and substantially increased bulk milk somatic cell counts in a dairy herd. During a herd visit, management and hygiene practices were evaluated and data from the milk quality control program were retrieved. Furthermore, quarter milk samples, mucosal swabs from farm cats and a dog, and swabs from the milking unit were aseptically collected. The samples were examined bacteriologically, and *S. canis* was identified using conventional phenotypic methods and matrix-assisted laser desorption ionization time-of-flight mass spectrometry. Genetic relationships between *S. canis* isolates were determined by multilocus sequence typing, revealing that all *S. canis* isolates shared the same sequence type, presenting a new combination of alleles for which a new number (ST55) was assigned. As the most likely source of intramammary infection, a farmyard cat was identified. The concurrent treatment of all positive cows and the improvement of management (no further access of carnivores to the barn) lead to positive results, including a decreased somatic cell count.

**Abstract:**

The present case report provides data on the phenotypic and genotypic properties of *S. canis* isolated from nine dairy cows with subclinical mastitis (SCC greater than 200,000 cells/mL in the quarter milk sample, no clinical signs) and from a cat living in the barn and reports the eradication of the pathogen from the herd with an automatic milking system. The isolates were identified using conventional bacteriology, matrix-assisted laser desorption ionization time-of-flight (MALDI-TOF) mass spectrometry (MS) and genetic relationships were investigated by multilocus sequence typing (MLST). Udder health management and hygiene instructions comprised the removal of the carnivores from the barn, strict monitoring of milking hygiene and techniques to avoid new infections via the milking robot, with simultaneous therapy for all infected cows. Phenotypic and genotypic properties of all isolates were identical. MLST revealed a unique sequence type (ST55) and a farmyard cat was identified as the most likely source of the *S. canis* infection in cows. The simultaneous treatment of all infected cows and management and hygiene improvements lead to a decreased SCC within four weeks.

## 1. Introduction

*Streptococcus (S.) canis* is a Gram-positive, catalase-, and CAMP-negative streptococcal bacterium belonging to Lancefield group G. Colonies on blood agar plates are non-mucoid and β-hemolytic [1]. It is commonly isolated from the oropharynx, genital mucosa and skin of asymptomatic and symptomatic carnivores, being identified as a cause of septicemia, otitis externa, pyometra, skin infections, necrotizing fasciitis, respiratory disease and urogenital infections [2,3,4]. Although *S. canis* is mainly an animal pathogen, an increasing number of human infections have been documented, with a wide range of clinical manifestations including bacteremia, urinary tract and bone infections, endocarditis and pneumonia [5,6,7,8,9,10]. In dairy cows, it represents a rare but contagious pathogen causing intramammary infection (IMI) of long duration [11,12]. Potential sources of infection in cows are cats and dogs with access to the barn [2,12]. Persistently infected cows usually spread the pathogen via the milking procedure, especially if udder health management is insufficient. This can result in massive outbreaks of clinical and subclinical mastitis [2]. It has been reported that isolates originating from these outbreaks in a herd are either a single clone or phylogenetically closely related [12]. Differences or similarities between *S. canis* isolates can be identified via phenotypic methods (e.g., biotyping based on lactose and trehalose fermentation) or by applying genotyping techniques such as random amplified polymorphic DNA (RAPD), macrorestriction profiling using pulsed-field gel electrophoresis (PFGE), *emm* and *scm* (genes encoding the M and M-like protein) typing and multilocus sequence typing (MLST) [6,10,12,13,14,15,16,17,18,19,20]. MLST is a standardized, housekeeping gene sequence-based technique offering unambiguous, high-resolution results in the form of allelic profile data and sequence type (ST) assignment, useful for analysis of population phylogeny [21,22]. MLST has proven valuable in linking *S. canis* STs to host species, geographic location, time of isolation, and disease patterns (https://pubmlst.org/scanis). It provides insights into the genetic diversity of *S. canis* as well as possible host adaptions of certain STs and their ability to cause disease [23]. A recent study using MLST revealed that certain *S. canis* lineages may be present in several hosts (human, wild animal species and companion animals), suggesting a broad distribution [18]. In contrast, Richards et al. (2012) reported significant differences between carnivore and bovine isolates, suggesting a possible host adaption of certain genotypes. Most of the strains isolated from cows with IMI were determined to be ST14, ST2 and ST1. ST1 was also the most frequent in all tested feline and canine isolates [24]. The two bovine isolates in the database (https://pubmlst.org/scanis/), belong to ST9 [23]. Recently ST9, ST27 and ST13 were found in dogs with ulcerative keratitis [25]. Among human *S. canis* isolates, ST9 and ST13 are the most prevalent [10]. Although mastitis outbreaks caused by *S. canis* are reported from all over the world [2,12,26,27], genotyping of these isolates has rarely been performed. Here, we describe the isolation of *S. canis* from milk samples taken from dairy cows affected by subclinical mastitis (SCC greater than 200,000 cells/mL in the quarter milk sample, no clinical signs [28]) from one herd as well as from a farmyard cat. All isolates were phenotypically characterized and genotyped using MLST. This revealed that the outbreak was caused by a single clone belonging to a novel sequence type (ST55). Furthermore, the eradication of the pathogen from the dairy herd with an automatic milking system is described.

## 2. Materials and Methods

### 2.1. Farm and Data Collection

In September 2019, milk samples from six dairy cows with a history of high somatic cell count (SCC) were submitted to the diagnostic laboratory of the University Clinic for Ruminants of the University of Veterinary Medicine, Vienna, Austria for bacteriological examination. The family owned dairy farm was located in Salzburg, Austria. The 59 lactating cows (51 Simmental, seven Holstein Friesian und one Pinzgauer) were housed in a free stall with rubber mat cubicles. The cows were fed grass silage, maize silage and hay ad libitum. Concentrates were fed in the milking robot. The farm participated in the national milk quality control program, including the recording of the milk yield, single animal and bulk milk SCC (BMSCC), milk fat and protein concentration. The average milk yield was 8400 kg per animal in 305 days. All cows were milked with the same automatic milking system (Voluntary Milking System, VMS, DeLaval, Tumba, Sweden) since 2012. Because *S. canis* was isolated in three cows with high SCC, a herd visit was performed to evaluate prevalence and clinical course of *S. canis* mastitis within the herd and to identify possible sources. During this visit, additional information on milking performance and milking hygiene practices were collected, and five consecutive milking’s were observed with the aim to document failures in functionality of the device (teat cleaning, teat finding, coverage of teat ends with disinfectant during teat spraying, flushing and disinfection of clusters between milking). The concentration of peracetic acid used for cluster disinfection was measured with commercially available test sticks (Quidee, Homberg, Germany). Results of former bacteriological examinations of milk samples were also obtained from the farmer.

### 2.2. Sample Collection

During the farm visit, quarter milk samples from all lactating cows (*n* = 59), except for the six cows already sampled the week before, were aseptically collected according to the method described by the National Mastitis Council [29]. Additionally, all lactating cows were tested using the Californian Mastitis Test (CMT) and a clinical examination of the udder of each cow was performed [30]. To find the potential source of *S. canis* infection and to determine possible modes of transmission, surface samples from the automatic milking system (teat cleaning cup (*n* = 2), teat liner (*n* = 2)) were collected. Therefore, swabs (steril cotton wool mounted on wooden rods) were inserted in the cleaning cup after cleaning and in the teat liners after milking an *S. canis* infected cow. As described by Munoz et al. (2007) [31], they were swabbed from the inside top to the opening of the cup/liner in a spiraling motion while rotating the swab. All seven farmyard cats and a dog were clinically examined and swabs from the nasal or pharyngeal mucosa of the carnivores (*n* = 8) were taken [30]. All swabs were inoculated and transported in trypticase soy broth (Merck, Darmstadt, Germany) to the diagnostic laboratory of the University Clinic for Ruminants of the University of Veterinary Medicine, Vienna, Austria.

### 2.3. Bacteriological Examination and SCC

Ten microliters of each milk sample and 100 µL of the environmental samples were, cultured on Columbia agar plate supplemented with 5% sheep blood (Oxoid Ltd., Basingstoke, UK) and incubated under aerobic conditions at 37 °C. Agar plates were evaluated for bacterial growth after 24 and 48 h. Colonies were identified based on colony morphology and hemolytic patterns on blood agar, as suggested by the National Mastitis Council [29]. Presumptive streptococci were confirmed by catalase test and further analyzed by Lancefield serogrouping (Streptococcal grouping kit; Oxoid Ltd., Basingstoke), cultivation on esculin hydrolysis agar (Oxoid Ltd., Basingstoke), and the CAMP test.

For further phenotypic characterization and identification at the species level, all catalase-negative, β-hemolytic isolates were examined by API^®^ RAPID ID 32 STREP (bioMérieux, Marcy l’Etoile, France) and MALDI-TOF mass spectrometry (Microflex LT Biotyper, Bruker Daltonics, Bremen, Germany). For the latter, single colonies were suspended in 300 μL HPLC-grade water and 900 μL of absolute ethanol, followed by centrifugation at 20,000× *g* for 5 min. Then, the supernatant was removed, and the pellet resuspended with 30 μL of 70% formic acid and 30 μL acetonitrile. After centrifugation at 20,000× *g* for 2 min, 1 μL of supernatant was spotted onto a MALDI target plate, air-dried, and overlaid with 1 μL α-cyano-4-hydroxycinnamic acid matrix solution. For the generation of mass spectra, the microflex LT Biotyper operating system (Bruker Daltonics, Bremen, Germany) was used. Data were analyzed using Bruker FlexControl 3.4 and MBT Compass Explorer 4.1 software (Bruker Daltonics, Bremen, Germany) with an integrated taxonomy database containing reference spectra. A logarithmic identification score was used to express the degree of spectral concordance and score values above 2.000 were considered to be acceptable for identification at the species level, according to the criteria proposed by the manufacturer.

Quarter milk SCC of all cows was determined using DeLaval Cell Counter (DeLaval, Tumba, Sweden).

### 2.4. Susceptibility Testing

For antimicrobial susceptibility testing, a Kirby–Bauer agar disk diffusion test was carried out in accordance with the CLSI (Clinical and Laboratory Standards Institute) and EUCAST (European Committee on Antimicrobial Susceptibility Testing) standards [32,33]. The following antimicrobials were tested: penicillin, ampicillin, clindamycin and cephalexin/kanamycin. As there are no zone diameter breakpoints for *S. canis*, those from *Streptococcus (S.) agalactiae* were used [32,33,34]. One *S. canis* isolate per cow and the positive cultured feline isolate were tested (*n* = 10).

### 2.5. Multilocus Sequence Typing

MLST was applied for genotypic characterization of nine bovine and one feline isolate, which were phenotypically identified as *S. canis*. MLST was performed on seven housekeeping genes (glucose kinase gene (gki), glutamine transport protein gene (gtr), glutamate racemase gene (murI), DNA mismatch repair protein gene (mutS), transketolase gene (recP), xanthinine phosphoribosyl transferase gene (xpt), and acetyl-CoA-acetyltransferase gene (yqiZ)), as previously described [14]. Sequences of each locus were assigned to allele numbers and their combination was utilized to define the ST using the public database at https://pubmlst.org/scanis [23]. For phylogenetic analysis, concatenated sequences of a representative of each distinct *S. canis* ST were obtained from the database and imported into the MEGA 6.06 software package (http://www.megasoftware.net/). Concatenated sequences with a total of 3134 positions were aligned by ClustalW and a phylogenetic tree was constructed using the neighbor-joining algorithm and Kimura two-parameter substitution model, and the topology was validated by bootstrapping (1000 replicates) [14].

## 3. Results

### 3.1. Animals and Herd Description

Evaluation of the milking revealed proper milking technique and hygiene. The farm used free cow traffic and the mean daily milking frequency was 2.4 milkings per cow per day. Milking intervals were on average eight hours. Visual evaluation of the milking procedure and assessment of cleanness and condition of teat skin confirmed the effectiveness of teat cleaning and teat cup attachment. Teat disinfection after milking was conducted automatically by spraying with an iodine teat disinfectant (DeLaval Prima, DeLaval, Tumba, Sweden). Observation of five consecutive milkings revealed that teat cleaning, attaching, adjusting and removal of the milking unit, teat disinfection by spraying with an iodine teat disinfectant (DeLaval Prima, DeLaval) and cluster disinfection were carried out properly. Concentration of the peracetic acid solution, as measured with test sticks was >500 ppm. Teat liners were changed every 2800 milkings. The farmer used electrical conductivity, milk color measurement and drops in milk yield for mastitis detection. All lactating cows were dried off using blanket dry cow therapy with Nafpenzal^®^ (180 mg penicillin G sodium, 100 mg dihydrostreptomycin and 100 mg nafcillin, Intervet, Kenilworth, NJ, USA) and teat sealers. In Table 1, the number of cows with subclinical (SCC threshold greater than 200,000, no clinical signs) and clinical mastitis (milk alterations) [28], as well as the course of the SCC of the composite milk samples (CMSCC) is stated. Because of the increasing CMSCC, the farm was consequently in danger of losing the milk market. The most recent CMSCC was 334,000 cells/mL. Overall, the herd did not have any apparent problems with hygiene practices, the cow’s environment was clean. However, visiting the farm revealed that seven cats and one dog were living (sleeping, feeding and defecating) in the barn.

### 3.2. Bacteriology, Evaluation of the Udder and SCC

The predominant pathogen found was *S. canis*, cultured from nine bovine milk samples and one feline mucosal swab. On Columbia agar containing 5% sheep blood, *S. canis* produced small (0.5–1.3 mm in diameter), white, non-mucoid colonies surrounded by a zone of complete hemolysis (β-hemolysis). They were Gram-positive, catalase-, esculin-, and CAMP-negative, and belonged to Lancefield group G. Using API^®^ RAPID ID 32 STREP (bioMérieux, Marcy l’Etoile, France) all isolates displayed identical biochemical properties corresponding to *S. canis* with a confidence value of 99% according to the manufacturer’s instruction (Profile: 7 6 1 1 6 0 4 1 1 1 0). The isolates were uniformly positive for arginine dihydrolase, β-glucosidase, β-galactosidase, alkaline phosphatase alanyl-phenylalanyl-proline arylamidase, α-galactosidase and methyl-βD glucopyranoside activity and produced acid from carbohydrates including ribose, lactose, saccharose and pullulan maltose but not for mannitol, sorbitol, trehalose, raffinose L-arabinose, D-arabitol, cyclodextrin, glycogen melibiose, melezitose, and tagatose. All isolates yielded negative reactions for β-glucuronidase, pyroglutamic acid arylamidase acetoine production (Voges Proskauer reaction), hippurate hydrolysis, n-acetyl beta-glucosaminidase, glycyl-tryptophane arylamidase, β-mannosidase and urease. Species identification was further confirmed by MALDI-TOF mass spectrometry, producing the highest log score values to S. *canis* reference spectra ranging between 2.19 and 2.35. No ambiguous results or misidentification as *S. dysgalactiae* ssp. *equisimilis* was observed using the extraction method described above (Appendix A).

In total, milk samples from 19 quarters in nine cows were positive for *S. canis*, nine cows for coagulase-negative staphylococci, four cows for *Staphylococcus (S.) aureus*, four cows for S. spp. and one cow for *E. coli* (Table 2). Two cows had two different pathogens in two different quarters (cow 23: *S. canis* and *S. aureus*, cow 58: *S. canis* and S. spp., Table 2). Clinical examination revealed that the cows infected with *S. canis*, had normal udders on adspection and palpation, one cow had clots in the milk of one quarter. Average SCC was 1,166,000 cells/mL in the *S. canis* positive cows (*n* = 9), 83,000 cells/mL for culture negative cows (*n* = 35) and 126,000 cells/mL for cows positive for pathogens other than *S. canis* (coagulases-negative staphylococci, *S. aureus*, *S. uberis*, S. spp.; *n* = 13). The SCC of *S. canis* infected quarters is listed in Table 2. All examined and tested carnivores were in a good clinical condition, except one cat presented with rhinitis. The pharyngeal mucosa swab from the latter was cultured as positive for *S. canis*. None of the swabs from the milking system were positive for *S. canis*.

### 3.3. Susceptibility Testing

Antibiotic susceptibility testing via the Kirby–Bauer agar disk diffusion test revealed that each isolate was susceptible to all tested antibiotics (penicillin, ampicillin, clindamycin and cephalexin/kanamycin). Clindamycin was tested as surrogate for lincomycin.

### 3.4. Multilocus Sequence Typing

By employing MLST, all nine bovine and the single feline isolates shared the same novel allelic profile (3, 2, 4, 9, 4, 7, 3) for which a new ST was assigned (ST55) [23]. In a phylogenetic tree constructed for concatenated sequences from all currently known *S. canis* STs, ST55 was grouped together with ST39, forming a clonal complex (SLV, single locus variant) with high bootstrap support of 95% (Figure 1).

### 3.5. Treatment and Interventions

The treatment comprised of immediate antibiotic therapy for all infected cows. As shown in Table 2, lactating cows (*n* = 6) were treated systemically with penethamate hydroiodide (Ingel-Mamycin^®^, 269.4 mg/mL, 10,000 IU/kg i.m., Boehringer Ingelheim, Ingelheim/Rhein, Germany) for four consecutive days. All infected quarters were treated intramammary with Benzylpenicillin procaine monohydrate (Vanaproc^®^, 333 mg/g, 3,000,000 IU, Vana, Vienna, Austria) once a day for five consecutive days (LCT, lactating cow treatment). Meloxicam (Metacam^®^ 20 mg/mL, 0.5 mg/kg s.c., Boehringer Ingelheim) was administered once when the treatment started. Blanket dry cow therapy (DCT) for all cows was recommended during the sanitation period. Cows close to dry off and cultured positive for *S. canis* (*n* = 3) were treated the same as the lactating cows and then dried off with Benestermycin^®^ (100 mg framycetinsulfat, 280 mg benethamin-penicillin, 100 mg penethamathydroiodid, Boehringer Ingelheim). Cows close to dry off with negative culture were treated in all four quarters with Benestermycin^®^ (*n* = 2, Table 2).

The farmer was advised to carefully monitor the cleaning and disinfection of the automatic milking unit including teat cups and the cleaning cup, as well as the post milking spraying with teat disinfectant. In order to prevent further infections, all pets had to leave the barn. Recommended control measures included control of treated cows two and five weeks after conclusion of therapy and of fresh cows immediately after calving by means of CMT and bacteriological examination. Cows which were nonresponsive to initial treatment had to be culled.

Unfortunately, contrary to our recommendations, no follow-up bacteriological examinations to monitor the treatment success and pathogen eradication in the herd were carried out. Shortly after our herd visit, the farmer carried out a recheck of the milking robot and milking system. The carnivores were banned from the barn, but with little success. After treatment, according to the farmer and the supervising veterinarian no new infection occurred. The next recorded CMSCC one month later was 139,000 cells/mL and to date (December 2020) it did not exceed 175,000 cells/mL. No further *S. canis* positive milk samples have been cultured so far.

## 4. Discussion

We report the first isolation and phenotypic and genotypic characterization of *S. canis* ST55 recovered from bovine milk samples and a feline pharyngeal swab during an outbreak causing subclinical mastitis. IMI caused by *S. canis* is rare. In ten years, out of 146,474 quarter milk samples which were sent for routine diagnostics to the diagnostic laboratory of the University Clinic for Ruminants of the University of Veterinary Medicine, Vienna, Austria, only 0.2% were cultured positive for *S. canis*. Nevertheless, outbreaks are reported from all over the world, while the epidemiological knowledge regarding the entry site of *S. canis* into bovine host, the course of disease within a herd and the transmission from cow to cow is still scarce [2,12,26,27].

In 2005, a cat was identified as a source of infection in a dairy herd in New York State. This cat had chronic sinusitis and spread *S. canis* while living in the barn. A more detailed route of infection could not be elucidated [2]. Just as in our study, the farmer never saw interactions between cats and cows (e.g., cats licking cow’s teat, cows licking the cat’s fur, cats licking parts of the milking unit). The farmer just reported that the cats and sometimes the dog were sleeping and defecating in the barn. The same was observed during our herd visit. Therefore, it could not exactly be determined how the infection started, whether the cows ate contaminated feed, excreted the pathogen and infected the quarters with feces, or whether the infection started at the udder because the teats were lying in bedding contaminated with feline excretions. On the other hand, it is common in Austria that carnivores live in the barn and are fed with unpasteurized milk. Perhaps the cat was infected through the cows, and the cows’ source of infection was a different one [2]. Further investigations on pathogen entrance and transmission are needed. It has been suggested that an *S. canis* outbreak in a dairy herd can occur without contact with carnivores. The introduction to a herd could occur because of *S. canis* on the milker’s skin [12]. The role of the milking unit as a source for pathogen transmission has been already reported [3,12,35]. In conventional milking systems, pathogen spread can be prevented by applying a constant milking order and ensuring intermediate disinfection of the milking cluster. As this farm works with an automatic milking system, it is a necessary to make sure that the milking technology works properly. Regular assessment, servicing and maintenance are especially important, as the efficient performance of the automatic milking, as well as the consistent operation of the system are essential for preventing pathogen transmission [36]. There are many machine related or cow related issues. Porous liners can become a vector for spreading infections, the wash up routine may be inadequate, the liner dimensions could be wrong, and cows may be restless or have suboptimal teat conformation. This, in particular, impairs the correct attachment of the milking cups. The concentration of the peracetic acid must be adjusted according the manufacturers manual and the milking unit must be sufficiently flushed and steamed between milkings. It has been reported that back flushing of artificially contaminated liners removed 98% of *S. agalactiae* [37]. None of the four swabs from the cleaning unit and milking unit cultured positive for *S. canis* in this survey. Although this is only a limited number of swabs, the result can be used as an indication that the automatic milking technology was working satisfactorily, which is in agreement with management observation during our herd visit.

All *S. canis* isolates were susceptible to all tested antibiotics (penicillin, ampicillin, clindamycin and cephalexin/kanamycin), which is in agreement with other findings in cattle [3,12]. However, intermediate resistance for erythromycin, tetracycline and gentamycin, and colistin sulfate as well as neomycin resistance in isolates from dairy herds affected by *S. canis* have been reported [26,27]. Similar results comprising resistance to neomycin, gentamycin as well as resistance or intermediate resistance to tetracycline has been found in human and carnivore isolates [7,14,37,38,39,40,41,42]. Intermediate susceptibility was seen for both enrofloxacin and orbifloxacin in carnivore isolates [43].

In 19 quarters, *S. canis* was the only pathogen. This is interesting as it has been reported that bovine *S. canis* positive quarter milk samples can be polymicrobial [12,26]. A human study revealed that *S. canis* infections were mostly associated with *S. aureus* (53.1%) or coagulase-negative staphylococci (12.5%) [8]. Although the latter isolates were not obtained from milk, this finding is similar to a previous study, which examined a total of 22 infected quarters while nine quarters had mixed infections with staphylococci [12]. Damage to the udder tissue or an increased susceptibility to pathogens could be a reason for this.

For the successful eradication of *S. canis* within a herd, carrying out similar management and control instructions to those applied in *S. agalactiae* eradication programs is advised. This is due to the fact that both *S. agalactiae* and *S. canis* have similar epidemiologic characteristics: high infectivity, low self-cure rate, high herd prevalence [2,12,43]. According to Tikofsky and Zadoks (2005) a cure is more likely when the treatment is performed in cows at drying off (87.5%), compared to cows treated during lactation (67%) and non-treated cows (9%) [2]. These authors treated most of the cows immediately, but some cows were not treated until dry off or remained in the herd untreated. This might lead to a reduction in prevalence but there is still a risk of exposure for negative herd mates to *S. canis* and reinfection can easily occur. This risk is increased when poor milking hygiene and management are additional factors. The main goal was to treat all infected cows immediately and improve the management at once to subsequently reduce the prevalence in order to avoid new and re-infections. It must be considered, that in a small herd it is easier to treat all cows at once and management changes can be implemented quicker. According to the farmer, the treatment was successful based on CMT, bulk milk SCC, automatic milking reports and the milk quality control program. Unfortunately, contrary to our advice, the farmer did not evaluate the therapy success two and five weeks after the end of therapy.

Different methods for the molecular characterization of isolates can be used. To our knowledge, MLST has not been used so far to detect the probable origin of an outbreak of subclinical IMI in a dairy herd. The advantage of MLST is that this kind of molecular typing allowed us to evaluate the genetic relationship between isolates obtained during an outbreak of subclinical mastitis to determine the source of infection. Furthermore, this method allows one to directly compare these isolates with other isolates from different hosts worldwide. According to the database (http://www.pubmlst.org/scanis/) [14,23], the allelic profiles of the isolates in this study were new and could not be assigned to any known ST, and therefore, they were given the new sequence type number 55. The phylogenetically most closely related to ST55 was found to be ST39 (3, 2, 4, 9, 4, 1, 3). Two isolates belonging to ST39 are now stored in the MLST database, both were isolated in 2016 in Germany, one from the respiratory tract of a cat and the other from a dog’s ear exudate. Further studies will expand knowledge about the diversity of *S. canis* MLST alleles and STs. Additional studies will offer insights into the occurrence, hosts, frequency, and pathogenic potential of these isolates. According to Pinho et al. (2013), this will allow for a better understanding of their veterinary importance [14].

## 5. Conclusions

This case report presents a dairy herd with an outbreak of subclinical mastitis caused by *Streptococcus canis* with highly increased somatic cell counts. By means of conventional bacteriology, multilocus sequence typing and analysis of herd management and hygiene practices, it was possible to identify the farmyard cat as the most likely source of infection. The immediate treatment of all positive cows at the same time, in addition to optimal management and hygiene measures led to results in minimal time, including a decreased SCC.

## Figures and Tables

**Figure 1 animals-11-00550-f001:**
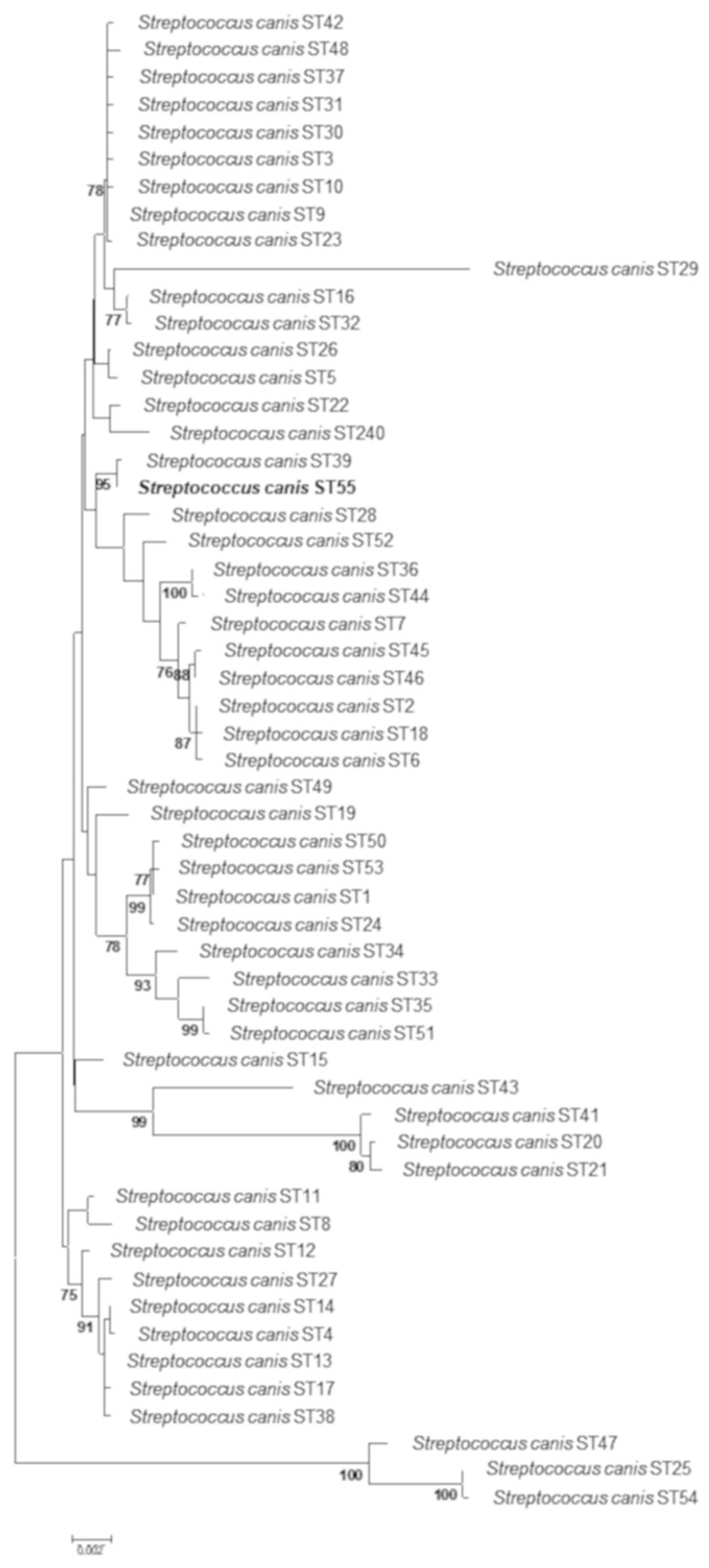
Neighbor-joining phylogenetic tree based on concatenated sequences of 7 housekeeping genes used in *S. canis* multilocus sequence typing (MLST) scheme, constructed from all currently known *S. canis* STs (*n* = 55). Numbers at nodes represent bootstrap confidence values (1000 replications). Only bootstrap values exceeding 75% are shown. The scale bar indicates the number of nucleotide substitutions per site.

**Table 1 animals-11-00550-t001:** The course of subclinical mastitis (SCC) in the affected herd (data from the national milk quality control program).

Cells/Animal/mL	2019-01-24 No. of Animals	2019-03-07 No. of Animals	2019-04-15 No. of Animals	2019-05-21 No. of Animals	2019-07-09 No. of Animals	2019-08-12 No. of Animals
<100,000	42	45	42	41	30	36
100,000–200,000	6	8	10	4	10	3
200,000–500,000	6	4	7	13	10	5
500,000–1,000,000	1	1	3	1	4	1
>1,000,000	3	4	2	2	5	6
Average cells/herd/mL	177	227	158	133	391	334

**Table 2 animals-11-00550-t002:** Results of the bacteriological and clinical examination of *S. canis* positive cows.

Cow ID	Quarter	CMT	Culture	SCC Cells/Quarter/mL and Milk Alteration	SCC Cells/Animal/mL	Days in Lactation	Intervention
12 *	FR	++	*++ S. canis*	ND	1,210,000	334	LCT
	RR	++	*+++ S. canis*	ND			
	FL	+	*+ S. canis*	ND			
	RL	+++	*+ S. canis*	ND			
7 *	RR	+	*+ S. canis*	ND	3,652,000	253	LCT
	RL	++	*++ S. canis*	ND			
33 *	FR	+++	*+ S. canis*	ND	1,730,000	348	DCT
	RR	+++	*+++ S. canis*	ND			
	FL	+++	*+ S. canis*	ND			
	RL	+++	*+++ S. canis*	ND			
5	FL	+++	*+++ S. canis*	3,507,000	257,000	326	DCT
10	FR	+++	*++ S. canis*	-	806,000	274	LCT
	RR	+++	*++ S. canis*	-			
	RL	+++	*++ S. canis*	947,000; clots			
23	FR	++	*+++ S. aureus*	30,000	336,000	264	LCT
	RL	+++	*++ S. canis*	2,937,000			
43	FR	+++	*++ S. canis*	4,957,000	400,000	243	LCT
52	FR	+++	*+++ S. canis*	214,000	1,780,000	275	DCT
	RR	+++	*++ S. canis*	3,757,000			
	RL	+++	*+ S.* spp.	968,000			
58	RL	+++	*++ S. canis*	1,132,000	324,000	18	LCT

* = part of the first survey, CMT = California mastitis test (+ = weekly positive score, apparent thickening, ++ = clearly positive score gel formation, +++ strongly positive score, firm gel formation in the center of the cup), Culture: + = 1 to 5 CFU, ++ = 6 to 10 CFU, +++ = more than 10 CFU). SCC = somatic cell count, FR = front right, RR = rear right, FL = front left, RL = rear left, ND = not done, DCT = dry cow treatment, LCT = lactating cow treatment.

## Data Availability

Data is contained within the article or Appendix A.

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
