# Peer review of "An Outbreak of Subclinical Mastitis in a Dairy Herd Caused by a Novel Streptococcus canis Sequence Type (ST55)"

_animals, 2021, doi:10.3390/ani11020550_

Round 1
Reviewer 1 Report
The revised paper reads very well and I have no further comments
Reviewer 2 Report
The authors justified why they did not do the PFGE and corrected the minor errors found.
Reviewer 3 Report
The authors have addressed my comments properly. I do not have further questions.
This manuscript is a resubmission of an earlier submission. The following is a list of the peer review reports and author responses from that submission.
Round 1
Reviewer 1 Report
General comments
It was a pleasure reading this interesting case report and well-written paper. I am not a veterinarian so I have focused on the sampling and machine part of the case. Material and methods are well described and conclusions are drawn on a sound basis.
Specific comments
Line 88-96: As a reader I was missing farm specific information here like number of cows and number of automatic milking machines but all this shows up in the Result chapter.
L103: Expand the explanation of how swabbing was carried out. Did you only swab inside the liners and cleaning cup or did you also swab the top of the cleaning cup and liners? Flushing will take care of the inside of the liners, but may not clean the top, which does come in direct contact with the udder basis.
L157: Rewrite to: All cows were milked with the same automatic milking system (Voluntary….
L163: Did you observe if every teat end was covered with the iodine teat disinfectant after spraying? The VMS use to be OK, but it is necessary to check it.
L165: Add concentration of peracetic acid. Add: after milking of each cow.
L227-239: How about the cat was that one treated as well?
L247: Really sad that you did not get the chance to do bacterial examination again.
L265-276: We would have loved to see the infection route. A nice discussion here and important that you point out that infection route between the cat and the cows could have been reversed.
Reviewer 2 Report
Major revision: I suggest changing Figure 1 for a similarity dendrogram of all S. canis strains obtained in the study by the pulsed Field Gel Electroforesis (PFGE) technique with the description of the ST of these strains in the same figure.
Minor revision:
- Change S. canis for S. canis in the Results, Discussion and check the entire manuscript sections.
- Results lines 196-208, and Discussion lines 304-327: all microorganism species names need to be written in italic.
Change St. aureus in Results and Discussion for S. aureus. St. aureus is incorrect.
Discussion, line 312: Change S.s agalactiae for S. agalactiae.
Author Response
Please see attachment,
Thank you

Reviewer 3 Report
General comments:
The case report manuscript by Eibl et al. reported an outbreak of subclinical mastitis in a dairy herd caused by a novel S. canis sequence type (ST55). Conventional bacteriology and MALDI-MS were used to identify isolates and MLST was used to investigate genetic relationships. Based on quarter milk samples, mucosal swabs from farm cats and a dog, and swabs from the milking unit, a unique sequence type (ST55) and a farm cat was identified as a potential source of ST55 infection in nine cows. The subsequent treatment, hygiene management, and removing the cat from farm lead to positive results such as reduced SCC. Overall, the manuscript is interesting and well written. However, several major points need to be addressed in the current version of manuscript. For example, the definition of subclinical mastitis is unclear. Based SCC in Table 1, several cows actually had clinical mastitis. The authors need to state if they also examined gram-negative bacteria or not? In addition, MALDI-TOF MS data is missing. Please see below for more details:
Major points:
- The definition of subclinical mastitis should be added to ‘Introduction’.
- L87-96: Did you collect other clinical data for the cows? For example, besides subclinical mastitis, whether other periparturient diseases were observed? Several other metabolic/infectious diseases could increase the likelihood of subclinical/clinical mastitis.
- L97-109: It is unclear when the milk samples were collected (e.g., the range of lactation date after post-calving, it seems like some cows were in dry-off period based on Table 2)? As a qualitative measurement for SCC in milk, CMT has relatively lower sensitivity and specificity. Why did not you use infrared spectroscopy to measure SCC?
- L116-121: Did you examine gram-negative bacteria?
- L122-125: More details need to be provided for MALDI-TOF MS procedures including sample preparation, m/z annotation, tandem MS/MS data, and data processing. Microflex from Bruker has relative low resolving power. The authors need to provide convincing data about how your targeted m/z’s were annotated and confirmed.
- L175: What cut-offs of SCC you used to define subclinical and clinical mastitis?
- L193-195: Where is MALDI-TOF MS data? Representative mass spectra (both MS and MS/MS) and reference spectra need to be provided in ‘Results’. The annotation procedures need to be clearly described.
- L197-198: Why did not you follow up on cows infected by St. aureus, S. spp. And E. coli?
Some minor points:
- Line (L) 39: Define SCC in the abstract.
- L 39: ‘a short period of time’: for how long? Exact time period should be provided.
- L98: How many lactating cows were used for milk sampling?
- L101-102: What criteria were used for clinical examination of the udder of each cow?
- L104: Why only two teat cleaning cup and two teat liner were used for collecting surface samples?
- L129: What are CLSI and EUCAST?
- L153: The diet information should be provided in supplementary files.
- L155-156: Did you compare milk yield, SCC, milk fat and protein between the nine sick cows versus healthy cows?
- L172: Double check the unit. Is it ‘cells/herd/mL’ or ‘cells/animal/mL’?
- L223-225: Caption for Figure 1 should more detailed for regular readers.
